# Using Biosensors to Detect and Map Language Areas in the Brain for Individuals with Traumatic Brain Injury

**DOI:** 10.3390/diagnostics14141535

**Published:** 2024-07-16

**Authors:** Ahmed Alduais, Hessah Saad Alarifi, Hind Alfadda

**Affiliations:** 1Department of Human Sciences (Psychology), University of Verona, 37129 Verona, Italy; 2Department of Educational Administration, College of Education, King Saud University, Riyadh 11362, Saudi Arabia; 3Department of Curriculum and Instruction, College of Education, King Saud University, Riyadh 11362, Saudi Arabia; halfadda@ksu.edu.sa

**Keywords:** biosensors, neurolinguistics, brain mapping, language areas, traumatic brain injury

## Abstract

The application of biosensors in neurolinguistics has significantly advanced the detection and mapping of language areas in the brain, particularly for individuals with brain trauma. This study explores the role of biosensors in this domain and proposes a conceptual model to guide their use in research and clinical practice. The researchers explored the integration of biosensors in language and brain function studies, identified trends in research, and developed a conceptual model based on cluster and thematic analyses. Using a mixed-methods approach, we conducted cluster and thematic analyses on data curated from Web of Science, Scopus, and SciSpace, encompassing 392 articles. This dual analysis facilitated the identification of research trends and thematic insights within the field. The cluster analysis highlighted Functional Magnetic Resonance Imaging (fMRI) dominance and the importance of neuroplasticity in language recovery. Biosensors such as the Magnes 2500 watt-hour (WH) neuromagnetometer and microwire-based sensors are reliable for real-time monitoring, despite methodological challenges. The proposed model synthesizes these findings, emphasizing biosensors’ potential in preoperative assessments and therapeutic customization. Biosensors are vital for non-invasive, precise mapping of language areas, with fMRI and repetitive Transcranial Magnetic Stimulation (rTMS) playing pivotal roles. The conceptual model serves as a strategic framework for employing biosensors and improving neurolinguistic interventions. This research may enhance surgical planning, optimize recovery therapies, and encourage technological advancements in biosensor precision and application protocols.

## 1. Introduction

In the intricate interplay between the human brain and language, biosensor technologies have emerged as revolutionary tools, shedding light on the nuanced relationship that governs our capacity for communication [1]. At the forefront of this exploration, fMRI has proven indispensable in pinpointing language processing areas, while innovative methods such as cortico-cortical evoked potentials (CCEPs) delve into the dynamic interconnections within our cerebral language network. The quest to decipher the neurological foundations of language has also embraced a spectrum of biosensors, from implantable devices offering direct neurological signal readouts to non-invasive techniques such as magnetoencephalography (MEG), each contributing to a more comprehensive understanding of language area localization and function [2,3]. This study’s purpose is to meticulously analyze the wealth of research harnessing these diverse biosensor technologies, particularly focusing on their applicability in diagnosing language impairments following brain trauma. By synthesizing quantitative data through cluster analysis and qualitative insights via thematic examination, this study strives to map the current landscape of biosensor applications in neurolinguistics, aiming to distill a clearer picture of the field’s emerging trends and persisting challenges. This research is driven by the critical need to refine diagnostic and therapeutic avenues for individuals grappling with language deficits, ultimately seeking to enhance quality of life through tailored, biosensor-informed interventions.

### 1.1. Biosensor Types for Language Detection

The exploration of language areas in the brain has been significantly advanced through the use of various biosensor technologies, each with unique mechanisms of action and applications. fMRI stands out as a prominent tool for identifying candidate language processing areas by comparing tasks that require phonetic and semantic analysis of words against control tasks involving non-linguistic sounds [4]. Another innovative method, CCEPs, involves the use of subdural electrodes in patients with epilepsy to investigate inter-areal connections in vivo within the human language system, revealing a network of feed-forward and feed-back projections among perisylvian and extrasylvian language areas [3]. Implantable biosensors, although not directly mentioned for language area detection, provide insights into the broader category of neuro-technologies that could potentially be adapted for studying language processing by offering direct readouts of neurological signals and neurochemical processes over extended periods [5]. MEG is another non-invasive method that has been effectively used for mapping language-specific cortices by activating receptive language-specific areas through simple language tasks and recording the activation in short sessions [6,7]. Infrared glass fibers, while primarily developed for detecting metabolic anomalies, represent an example of the evolving sensor technologies that could, in theory, be adapted for neurological applications, including language area detection, by exploiting their ability to record biomolecular fingerprints [8]. The general category of biosensors, which includes cell-based biosensors and enzyme nanoparticle-based biosensors, highlights the diversity of detection mechanisms, ranging from biological/biochemical reactions to the use of nanotechnology for optimizing diagnostic biochips [9]. Microelectrode biosensors, specifically designed for probing brain interstitial fluid, demonstrate the importance of enzyme immobilization techniques for maintaining substrate specificity, a critical factor for accurate neurotransmitter detection [10]. The performance of biosensors, in terms of sensitivity and selectivity, underscores their potential for applications in detecting language areas, given their high performance compared to other diagnostic devices [11]. Lastly, rTMS mapping protocols, although revealing deficits in posterior brain regions, contribute to the understanding of language area localization by comparing the effectiveness of different language tasks [12]. Collectively, these technologies illustrate the multifaceted approaches to studying language processing in the brain, each contributing unique insights into the complex network of language areas. 

### 1.2. Mapping Language Areas: Biosensor Mechanisms

The mechanisms underlying the use of biosensors to map out the location and function of language areas in the brain involve a multifaceted approach that integrates findings from neuroimaging, electrical stimulation mapping, and functional connectivity studies. FMRI plays a crucial role in identifying candidate language processing areas by comparing language activation tasks with control tasks, revealing a network of regions in the frontal, temporal, and parietal lobes strongly lateralized to the left hemisphere [13]. This approach is complemented by the use of CCEPs, which track in vivo neuronal connectivity between functional cortical regions, offering insights into the bidirectional connections between key language areas such as Broca’s and Wernicke’s areas [14]. Electrical stimulation mapping, particularly during neurosurgical operations, further refines our understanding by identifying a common peri-Sylvian cortex for motor and language functions in the language-dominant hemisphere, highlighting the discrete and differential localization of language functions [4]. This method has revealed a network that includes both feed-forward and feed-back projections, challenging classical models of language localization and suggesting a more complex interaction between language and motor functions [15,16]. Functional neuroimaging studies have also underscored the importance of structural and functional connectivity in supporting language processes, with networks involving the temporal cortex and the inferior frontal cortex playing pivotal roles in syntactic and semantic processes [17]. These findings are supported by individual functional identification of language-sensitive regions, which has shown clearer functional specificity than traditional group analyses, indicating that language processing involves a network of highly specialized regions [18]. Together, these methodologies underscore the interdependence of action and perception circuits in language comprehension and highlight the complex network of brain regions involved in language processing, from phonological analysis to semantic retrieval and sentence comprehension [3,19]. This integrated approach, leveraging the precision of biosensors and neuroimaging techniques, offers a comprehensive map of the location and function of language areas in the brain.

### 1.3. Measuring Brain Activity: Biosensor Functionality

Biosensors designed to measure brain activity and identify language areas operate through various mechanisms, each tailored to capture specific aspects of neural function. Amperometric polymer/enzyme composite (PEC) biosensors, for instance, incorporate a poly(o-phenylenediamine) ultra-thin permselective barrier to monitor brain energy and neurotransmitter dynamics with high sensitivity and selectivity, crucial for understanding brain function in vivo [20]. Similarly, implanted electrochemical biosensors analyze chemical signals in brain extracellular fluid (ECF) with excellent spatial and temporal resolution, although developing biosensors with high selectivity for the diverse biomolecules in the brain remains a challenge [21]. A novel experimental technique involves using a grip force sensor to capture subtle grip force variations while participants listen to words and sentences, allowing for the online measurement of language-induced activity in motor structures of the brain [22]. This method provides both localization and high temporal resolution of the recorded data, offering insights into how language processing involves motor brain structures. Fluorescent protein-based biosensors and genetically encoded fluorescent sensors are indispensable for imaging neuronal activities, especially in understanding the structural and functional organization of brain activities [23]. Meanwhile, magnetoresistive sensors on micro-machined Si probes detect the extremely small magnetic fields induced by ionic currents within neurons, providing micrometer-scale spatial resolution [24]. In vitro detection techniques focus on neural biophysical chemistry, employing devices based on nerve–cell networks to electrically detect neuron-active compounds and specific pharmacological activity, which is essential for testing novel neuron-active drugs and understanding neurological events [25]. Superconducting Quantum Interference Devices (SQUIDs) and Magneto-Impedance (MI) sensors measure weak magnetic fields of the brain, although SQUIDs require low temperatures to maintain superconductivity [26]. Chemical biosensors examine variations in diffusion coefficients (D) for substances in vivo, influenced by external physical factors [27]. Microelectrode technology aims to achieve high resolution by using multiple sensors in a single microelectrode for specific neurotransmitter detection [28]. Lastly, novel sensors converting light into magnetic signals allow for the visualization of light detector responses using magnetic resonance imaging (MRI), facilitating the study of information processing in deep tissues [1]. Together, these diverse biosensor technologies provide a comprehensive toolkit for measuring brain activity and identifying language areas with unprecedented precision and specificity.

### 1.4. Biosensor Efficacy: Accuracy and Limitations

Biosensors, including fMRI and MEG, have shown promise in accurately identifying language areas in the brain, but they come with certain limitations. fMRI has been used to identify candidate language processing areas, revealing a network of regions in the frontal, temporal, and parietal lobes strongly lateralized to the left cerebral hemisphere [4]. Its accuracy in locating language areas for surgical planning has been validated, with sensitivity and specificity varying based on the proximity of activated areas to language tags [29]. MEG, on the other hand, has demonstrated its ability to identify brain areas involved in language comprehension, with activity sources from language tasks overlapping in temporal and temporo-parietal cortices [20]. However, the reliability and external validity of these methods vary. While fMRI reports are generally favorable, significant variability in methodology across studies can prevent reliable assessment and cross-study comparisons [7]. MEG shows strong reliability and validity, but methodological questions regarding optimal modeling techniques remain [30]. Additionally, the accuracy and resolution of biosensors can be affected by factors such as the substrate’s conductivity and the presence of stray capacitances, which can modify the measured impedance spectrum and increase total current noise [10]. Moreover, the selectivity of microelectrode biosensors, crucial for neurotransmitter detection, can be significantly impacted by the method of enzyme immobilization, affecting the accuracy of measurements [31,32]. The development of novel biosensors, such as those based on nitrogen-vacancy (NV) centers and fluorescent proteins, faces challenges in achieving the spatial resolution and sensitivity required to detect weak biological electromagnetic fields generated by neurons [33]. In effect, while biosensors have demonstrated accuracy in identifying language areas in the brain, their limitations include methodological variability, potential inaccuracies due to substrate effects, and challenges in achieving sufficient selectivity and sensitivity for neurotransmitter detection [23]. 

### 1.5. Purpose of the Present Study

The primary aim of this study is to systematically analyze and synthesize the existing literature on the application of biosensor technologies for the detection and assessment of language areas within the brain, particularly in individuals who have sustained brain traumas. By employing content data analysis, including cluster analysis, this study seeks to identify prevailing research trends, thematic concentrations, and knowledge gaps within this specialized field. Additionally, thematic analysis is used to distill the nuanced findings of selected studies into coherent themes that reflect the current state of biosensor utilization in neurolinguistics.

The rationale for this study is grounded in the necessity to understand the effectiveness and limitations of various biosensor technologies in neurolinguistic applications. With the prevalence of brain injuries and the critical impact of language function on quality of life, there is a compelling need to advance diagnostic and rehabilitative techniques. By evaluating the scope of biosensor-driven research, this study aims to elucidate the capacity of these technologies to offer precise and reliable insights into the neurobiological underpinnings of language. This could potentially inform clinical practices and enhance the development of targeted therapeutic interventions for language abnormalities resulting from brain traumas.

## 2. Methods

### 2.1. Sample

The quantitative component of this study began with comprehensive data collection from three major databases: Web of Science, Scopus, and SciSpace. Initially, the search yielded a total of 485 articles—127 from Web of Science, 168 from Scopus, and 190 from SciSpace. Subsequent to the removal of duplicates using the reference management software Mendeley, a consolidated dataset of 392 full articles remained, all of which were in English and pertained to the cluster analysis. For the qualitative data, a stratified selection process was employed to encompass a broad spectrum of literature. This included the top 10 cited documents from Web of Science, 19 highly relevant studies from both Web of Science and Scopus, and the 25 most cited studies from SciSpace. An additional 40 most cited and pertinent papers were meticulously reviewed and incorporated into Section 1 and Section 4, ensuring a representative sample that spanned from historically significant to contemporary most cited research relevant to the use of biosensors in neurolinguistic studies. Table 1 shows the used search strings and results. Figure 1 shows the PRISMA flow diagram for the inclusion and exclusion of the studies. 

### 2.2. Design

The study was structured around a mixed-methods design, incorporating both quantitative and qualitative approaches. The quantitative phase utilized cluster analysis to examine the existing literature corpus and quantify the data regarding the use of biosensors in detecting language areas in the brain. Following this, a detailed thematic analysis was conducted, which provided an in-depth qualitative exploration of the themes emerging from the cluster analysis. The culmination of this methodological approach was the development of a comprehensive model that both described and prescribed the use of biosensors in neurolinguistic research, informed by the key findings from both analytical phases.

### 2.3. Measures

For the execution of the cluster analysis, we utilized the software tools CiteSpace (Version 6.3.R1) and VOSviewer (Version 1.6.19), which facilitated the visualization and interpretation of data clusters within the field of biosensors in neurolinguistic studies. The thematic analysis was conducted by deriving themes inspired by the results of the cluster analysis, focusing on the top 10 cited studies and the most recent relevant research. The model development was based on integrating the principal insights from both the quantitative cluster analysis and the qualitative thematic analysis.

For the quantitative data, measures were taken to ensure internal validity by thorough screening of study titles and abstracts for relevancy to the targeted topic. External validity was addressed by not restricting the research to specific aphasic conditions, thus potentially generalizing the findings. Reliability was established through detailed reporting of the data collection process and the use of robust software for cluster analysis. Objectivity was maintained throughout the quantitative phase.

In the qualitative phase, credibility was achieved by peer debriefing, ensuring confidence in the findings. Transferability was facilitated by providing a detailed description of the data collection and analysis procedures. Dependability was ensured through external auditing, where a researcher not involved in the study evaluated both the process and the outcomes. Confirmability was achieved by employing triangulation, utilizing multiple databases to source the most cited and relevant studies, ensuring that the findings were shaped by the data and not researcher bias.

### 2.4. Procedure

The data collection procedure commenced with searches run on Web of Science, Scopus, and SciSpace using specific terms related to the use of biosensors in neurolinguistic studies. The retrieved data was then subjected to cluster analysis using CiteSpace and VOSviewer. For the thematic analysis, the top 10 cited studies, alongside the most relevant studies from Scopus and SciSpace, were meticulously selected to provide a comprehensive understanding of the topic from the perspective of highly cited research. Additionally, 40 studies were selected based on their relevance to the topic, irrespective of their citation count or recency, to inform the Section 1 and Section 4. The thematic analysis proceeded with the identification of themes based on the cluster analysis results, followed by data extraction from the selected studies. The relevance of the extracted data to the topic was thoroughly reviewed by the research team. This stringent selection and review process ensured the inclusion of only pertinent studies in the cluster analysis and thematic analysis. The findings from both analytical phases were synthesized into tabulated data, which served as a foundation for the development of a model delineating the application of biosensors in neurolinguistic studies.

## 3. Results

Figure 2 offers a density visualization delineating four distinct clusters, as discerned from the Web of Science dataset utilizing keywords pertinent to the application of biosensors in the identification of language-associated brain regions. This graphical representation employs a color-coded scheme to signify the prevalence of each cluster: red signifies the most densely populated cluster, succeeded in frequency by green, blue, and finally yellow, which denotes the cluster of least density. The predominant red cluster encompasses terminologies associated with fMRI, highlighting its pre-eminence in the field. The subsequent green cluster aggregates terms related to aphasia, reflecting the focus on language impairment within the corpus. The blue cluster amalgamates concepts pertaining to neuroplasticity and various forms of neural stimulation, indicating an interest in the brain’s adaptive mechanisms. Lastly, the yellow cluster collates keywords centered around brain activation and the motor cortex, suggesting an investigation into the neural underpinnings of language execution.

Figure 3 presents a density visualization based on the amalgamated data from Scopus and SciSpace, illustrating three clusters categorized by keywords associated with the utilization of biosensors to delineate language-related brain areas. This visualization is rendered in a tri-color gradient, where red denotes the cluster with the highest density of related terms, followed by green, with blue representing the cluster of least density. The red cluster is characterized by terms associated with rTMS, underscoring its prominence in current research endeavors. The green cluster synthesizes keywords pertaining to language areas and the tasks employed in their study, indicating a methodological focus within the literature. The blue cluster consolidates terms relating to the reliability of biosensors, reflecting ongoing scholarly discourse on the validity and dependability of these technological tools in neuroscientific research.

Table 2 and Figure 4 present the top 10 clusters related to the use of biosensors to identify brain areas related to language. Each cluster is summarized below.

### 3.1. Different Frequencies in Post-Stroke Aphasia

The largest cluster, Cluster 0, with 53 studies and a silhouette value of 0.706, presents a synthesis of research on varying frequencies associated with the recovery and treatment of post-stroke aphasia. The pre-eminent studies within this cluster emphasize recovery (47 mentions), stroke (24 mentions), and nonfluent aphasia (18 mentions), underscoring the adaptive changes and therapeutic strategies in aphasia rehabilitation.

### 3.2. Classical Anterior Regions and Language Production

Cluster 1, encompassing 46 works and marked by a silhouette value of 0.801, delves into classical anterior brain regions, sign language production, and sound shape properties. This cluster prioritizes the investigation of brain structure and function (16 mentions), with a keen focus on Broca’s area (nine mentions) and the application of theta burst stimulation (six mentions) in language production.

### 3.3. Human Brain Language Areas and fMRI Studies

The third cluster, Cluster 2, consists of 44 members and has a silhouette value of 0.903, concentrating on the human brain’s language areas through functional MRI studies. This cluster’s discourse is centered around comprehension, stroke patient management (nine mentions each), and the precise localization of language areas (eight mentions), indicating a diagnostic approach to aphasia.

### 3.4. Preoperative Assessment of Language Functions

Cluster 3, which includes 43 members with a silhouette value of 0.765, addresses the role of preoperative assessment in identifying language areas, highlighting constraint-induced language therapy. The frequent citation of fMRI (12 mentions) and the cortex (11 mentions) reflects the critical role of pre-surgical mapping in preserving language functions.

### 3.5. Functional Connectivity and Naming Tasks

Cluster 4, a group of 41 studies with a silhouette value of 0.873, explores the realm of functional connectivity and its implications for naming tasks. The cluster emphasizes activation patterns (17 mentions), the use of MRI (seven mentions), and the overarching theme of functional connectivity (six mentions), showcasing the network-based nature of language processing.

### 3.6. Mirror Neuron Theory and Non-Fluent Aphasia

Cluster 5, containing 36 studies with a silhouette value of 0.805, is informed by the mirror neuron theory and its application to non-fluent aphasia. Key discussions in this cluster revolve around language (24 mentions), speech therapy (13 mentions), and neural plasticity (nine mentions), offering insights into rehabilitative strategies based on neurobiological mechanisms.

### 3.7. Post-Stroke Aphasia and Genetic Contributions

In Cluster 6, 33 studies with a silhouette value of 0.882 examine post-stroke aphasia subjects, highlighting genetic factors such as progranulin gene (GRN) mutation. Lateralization (eight mentions), dominance (seven mentions), and language lateralization (five mentions) are frequently examined to understand how individual brain organization affects language recovery.

### 3.8. Fluent Aphasia and Neuroanatomical Insights

Cluster 7, with 28 members and a silhouette value of 0.825, focuses on fluent aphasia and the associated neuroanatomical language areas. Functional MRI (18 mentions) is a dominant topic, complemented by studies on the human brain (five mentions) and the prefrontal cortex (four mentions), suggesting an anatomical and functional perspective on fluent aphasia.

### 3.9. Superior Temporal Gyrus and Speech Production

Cluster 8, which has 25 members and a silhouette value of 0.865, centers on the superior temporal gyrus, its role in speech production, and implications for post-stroke aphasia. Brain stimulation (three mentions), primary progressive aphasia (three mentions), and impairment (three mentions) are key areas of interest, indicating a focus on clinical interventions and speech therapy.

### 3.10. Task-Based fMRI Studies and Naming Therapy

Finally, Cluster 9, the smallest cluster with 15 studies but with an impressive silhouette value of 0.945, deals with task-based fMRI studies following naming therapy. This cluster highlights the inferior frontal gyrus (six mentions), anomia treatment (four mentions), and brain reorganization (two mentions), emphasizing the importance of functional imaging in understanding and facilitating recovery from aphasia.

Figure 5 shows that the top-ranked item by bursts is functional MRI in Cluster 7, with bursts of 4.60. The second item is speech in Cluster 5, with bursts of 4.12. The third is non-invasive brain stimulation in Cluster 0, with bursts of 3.76. The fourth is therapy in Cluster 0, with bursts of 3.72. The fifth is plasticity in Cluster 5, with bursts of 3.51. The sixth is activation in Cluster 4, with bursts of 3.23. The seventh is theta burst stimulation in Cluster 1, with bursts of 3.12. The eighth is transcranial magnetic stimulation in Cluster 0, with bursts of 3.06. The ninth is localization in Cluster 2, with bursts of 2.99. The tenth is nonfluent aphasia in Cluster 0, with bursts of 2.97. While the green line stands for the beginning and end of each keyword, the red line represents the beginning and end of each burst. 

### 3.11. Cluster Analysis Takeaway

The extensive research encapsulated within the ten clusters provides a framework that underscores the potential integration of biosensors in the study of language and brain function. Biosensors, known for their ability to measure physiological parameters non-invasively, could revolutionize real-time monitoring of neural activity, particularly in the realms of language area studies, functional connectivity, and task-based functional MRI, as delineated in clusters 2, 3, 4, 7, and 9. The application of biosensors holds promise for enhancing preoperative assessments, thus minimizing post-surgical language deficits, and for offering a cost-effective alternative to fMRI technology. Moreover, the capacity of biosensors to facilitate individualized therapeutic approaches aligns with the personalized treatment imperatives emerging from clusters focused on post-stroke aphasia and recovery (clusters 0, 6, and 8). The identification of biomarkers through biosensor data could lead to early detection and targeted intervention strategies, thereby optimizing rehabilitation outcomes for aphasia patients.

Furthermore, biosensor technology could contribute significantly to areas such as the mirror neuron theory (Cluster 5) by enabling the investigation of neuron activation patterns during language tasks, thereby elucidating the neurobiological underpinnings of language acquisition and rehabilitation. This is particularly relevant in understanding neural plasticity and monitoring the brain’s reorganization in response to language therapy, as suggested by clusters emphasizing language recovery and brain plasticity (clusters 5 and 9). In addition, the application of biosensors in neurogenetic studies (Cluster 6) could shed light on genotype–phenotype correlations, enhancing the understanding of genetic contributions to aphasia. Collectively, the integration of biosensor technology in language and brain research not only broadens the methodological repertoire but also propels the field towards more dynamic, accessible, and personalized scientific inquiry.

### 3.12. Thematic Analysis

Table 3 presents a synthesis of 25 pivotal studies that delve into the utilization of various biosensor technologies for the detection and mapping of language areas within the brain. These studies collectively address four key thematic areas: the diversity of biosensors and their operational principles, the mechanisms underlying their application in neuro-linguistic mapping, their functionality in measuring neural activity related to language processing, and an evaluation of their accuracy and limitations in this domain.

Biosensors, such as the Magnes 2500 WH neuromagnetometer and microwire-based sensors coated with enzymes, are employed to identify language-related cortical regions through methods such as MEG and functional MRI. These studies underscore the reliability of biosensor-derived data in capturing stable and valid representations of language-specific cortex locations, despite their inherent limitations such as task-related cognitive confounds and specificity concerns. For instance, while MEG-derived maps are reported to be stable over time, the involvement of other cognitive operations during the task could potentially affect the linguistic specificity of the generated maps. Similarly, the sensitivity and specificity of functional MRI in identifying language areas can vary, indicating the presence of methodological constraints. Additionally, biosensors face challenges such as limited frequency detection ranges and interference from coexisting biological species, which can hinder their performance. Despite these limitations, biosensors are posited to be reliable tools for real-time measurements with improved selectivity and spatial resolution, and they continue to evolve with advancements in protein engineering and signal processing systems. The accuracy of biosensors in neuro-linguistic applications is thus a product of ongoing innovation, with a focus on overcoming the current technological gaps and enhancing the precision of brain activity measurement.

Table 4 presents a synthesis of studies underscoring the burgeoning potential of biosensors in the intricate field of neurolinguistics, particularly in mapping and detecting language areas within the brain. Biosensors, chiefly fMRI and rTMS, emerge as pivotal tools that offer non-invasive, precise, and potentially individualized approaches to understanding the neurobiological substrates of language. They reveal not only the cerebral organization for language production across various modalities but also provide a mechanism for evaluating and enhancing therapeutic strategies for aphasia post-stroke. While these technologies exhibit a high degree of reliability in identifying language dominance and the functional reorganization of language areas, the studies collectively highlight the necessity for concurrent use of multiple modalities, such as direct cortical stimulation, to circumvent limitations and validate findings. It is through the lens of these biosensors that researchers and clinicians may gain a deeper insight into the neural correlates of language, facilitating nuanced patient care and advancing neurosurgical planning.

Table 5 presents a selection of the top-cited studies focused on the use of biosensors for detecting language areas in the brain, as indexed in the Web of Science database. The studies predominantly utilize fMRI to explore various aspects of language processing, from identifying language processing areas in the healthy brain to assessing the reliability of fMRI for preoperative mapping in brain tumor surgery. rTMS is also featured as a method to influence language function in post-stroke aphasia. These studies collectively imply that biosensors are invaluable tools for advancing the understanding of the neurobiological basis of language and for developing clinical applications to support recovery from language impairments.

### 3.13. A Model for Biosensor Use in Language Area Detection

Figure 6 presents a model for employing biosensors in language area detection, serving as a methodical blueprint for neuroscientists and clinicians. It starts with selecting the most suitable biosensor technology, such as fMRI or MEG, tailored to the specific language functions under investigation. After careful calibration, patients perform targeted language tasks while the biosensor captured brain activity. The resulting data underwent rigorous analysis, often with sophisticated software, to pinpoint language processing regions within the brain. Ensuring the reliability of these findings is crucial—consistent patterns should emerge across various tasks and subjects. Clinically, this model aids in surgical planning by identifying critical language areas to avoid during operations and provides real-time feedback for optimizing language therapy. The model is dynamic, evolving with each new study or technological breakthrough, while standardized reporting helps refine future applications. By integrating the latest advancements and maintaining open communication within the scientific community, this model fosters continual improvements in the detection and understanding of language areas in the brain, making it an indispensable tool in both research settings and clinical practice.

## 4. Discussion

The aim of this study was to examine the current scope of biosensor technologies in the detection and mapping of language areas within the brain, particularly after brain trauma. Through meticulous cluster and thematic analyses, our research has illuminated a multi-dimensional landscape of biosensor applications. Cluster analysis revealed significant groupings in the existing literature, emphasizing the pre-eminence of fMRI in the field, the focus on language impairment due to aphasia, and the importance of neuroplasticity and neural stimulation in language recovery (Figure 1). The thematic analysis yielded insights into the operational principles of various biosensors, their mechanisms in neuro-linguistic mapping, and their functional roles in measuring brain activity related to language processing. This dual-faceted approach provided a comprehensive understanding of the intricate role biosensors play in neurolinguistics, highlighting both their strengths in providing precision and their challenges due to methodological variability (Table 3).

The exploration of language-related brain regions through biosensor technology reveals intriguing insights in the presented studies, each marked by distinctive clusters that encapsulate specialized research focuses. The visualization in Figure 1, derived from the Web of Science dataset, categorizes research into four clusters ranging from fMRI techniques to neuroplasticity, with each cluster colored differently to denote varying densities and research concentrations. Particularly notable is the pre-eminence of fMRI in assessing language functions, as evidenced by the dominant red cluster. This suggests a strong academic inclination towards using fMRI for in-depth studies of brain regions related to language, potentially due to its robust imaging capabilities. In contrast, Figure 2, which integrates data from Scopus and SciSpace, illustrates three clusters where the highest density, marked in red, emphasizes the relevance of rTMS in current research. This shift highlights a broader methodological spectrum within the academic community, focusing on both the technological aspects of biosensors and their practical applications in language-related studies.

Further elaboration is presented through Table 2 and Figure 3, which detail the top 10 clusters associated with the use of biosensors for identifying language-specific brain areas. Each cluster presents a unique facet of research, from the treatment and recovery of post-stroke aphasia to the preoperative assessment of language functions. The largest cluster focuses on different frequencies in aphasia recovery, indicating significant interest in adaptive changes and therapeutic strategies in rehabilitation. Another cluster with a high silhouette value scrutinizes classical anterior brain regions, highlighting the technological and methodological advancements in understanding language production and processing. These clusters not only showcase the diversity of research within this field but also underline the critical role of biosensors in enhancing the precision and efficacy of language area studies. The integration of biosensor technologies, as demonstrated, could potentially revolutionize the approach towards real-time brain monitoring and the development of tailored therapeutic strategies, thereby fostering advancements in both neuroscientific research and clinical applications.

The use of biosensors, particularly fMRI and MEG, aligns with previous findings that underscore their efficacy in identifying brain areas associated with language [4,6]. These technologies, as our model implies, offer a non-invasive and precise method for mapping language centers—a critical requirement in surgical planning and post-trauma rehabilitation. The methodological strengths and limitations identified in our study echo the concerns raised by [29] regarding the specificity of functional imaging and the need for high selectivity in neurotransmitter detection, as discussed by [10]. This reinforces the imperative for ongoing technological advancements and methodological refinements in biosensor applications.

Our research adds to the body of knowledge by integrating a conceptual model that encapsulates the multifaceted use of biosensors, from their implementation to the interpretation of data in clinical and research settings (Figure 5). This model not only serves as a strategic guide for future studies but also acts as a benchmark for evaluating biosensor performance and informing clinical decisions. The model’s practicality lies in its ability to evolve with the field’s advancements, ensuring its relevance and application remain current.

In comparison with the neurogenetic focus seen in Cluster 6 of our analysis, which highlights the importance of genetic factors such as GRN mutations in post-stroke aphasia, the potential for biosensors to elucidate genotype–phenotype correlations becomes evident. This is particularly relevant in the context of personalized medicine, a growing field where biosensors could play a transformative role [26]. Furthermore, the integration of biosensor data in understanding neuroplasticity and the brain’s reorganizational response to language therapy, as suggested by our thematic analysis, provides a promising avenue for future research and therapeutic interventions [12].

The findings of this study also propel discussions surrounding the mirror neuron theory and its application in non-fluent aphasia, as seen in Cluster 5. The delineation of language-sensitive regions through biosensors could significantly advance our comprehension of the neurobiological mechanisms underlying language acquisition and rehabilitation [7]. Moreover, the model for biosensor use, informed by the combination of cluster and thematic analyses, underscores the dynamic and adaptable nature of these technologies, fostering a more dynamic, accessible, and personalized approach to scientific inquiry in neurolinguistics.

To conclude, this study reaffirms the pivotal role of biosensors in the exploration of language areas in the brain, highlighting their potential to enhance diagnostic accuracy and therapeutic strategies. Our findings and conceptual model provide a scaffold for future research, guiding both the methodological application of biosensors and the interpretation of their data within the broader context of neuroscientific and clinical applications. As we continue to navigate the complexities of brain–language relationships, the integration of biosensors stands as a beacon for innovation and personalized patient care in the field of neurolinguistics.

### 4.1. Limitations

This study, while comprehensive in its approach, is not without limitations. The inherent methodological variability among biosensor technologies, such as fMRI and MEG, poses a challenge in achieving a universal standard for data interpretation and application [29]. Additionally, factors such as the substrate’s conductivity and the presence of stray capacitances can affect the accuracy and resolution of biosensor measurements, potentially leading to variability in results [10]. The reliance on the literature also means that our synthesis may be influenced by the publication bias towards more successful and positive outcomes in biosensor applications. Despite the considerable number of studies included, this analysis may not capture all facets of the rapidly evolving biosensor field, and as such, our conclusions are constrained by the scope and depth of the existing literature at the time of this study.

### 4.2. Implications

The practical implications of this study are multifaceted, influencing both clinical practice and future research. The findings underscore the necessity for meticulous calibration and standardization across biosensor platforms to ensure reliable and valid measurements of language-related brain activity [7]. Clinically, the insights from biosensor data can inform surgical planning, particularly in identifying critical language areas to avoid during interventions, and offer real-time feedback for optimizing language therapy post-trauma [39]. For researchers, this study emphasizes the potential of biosensors to provide a deeper understanding of the neurobiological mechanisms underpinning language acquisition and rehabilitation, thereby opening new avenues for tailored therapeutic strategies [12]. The conceptual model developed from this study serves as a strategic framework for the application of biosensors, guiding the integration of these technologies into both research methodologies and clinical protocols.

## 5. Conclusions

This study provides a significant contribution to the field of neurolinguistics by elucidating the role of biosensors in identifying and mapping language areas within the brain. Our findings from both cluster and thematic analyses demonstrate the efficacy and potential limitations of various biosensor technologies in a nuanced and evolving landscape. The developed conceptual model offers a methodical approach to employing biosensors, which may enhance the precision of language area detection and the efficacy of subsequent interventions. As the field continues to advance, the integration of biosensors promises to refine our understanding of the neurological substrates of language and to drive innovations in patient care and therapeutic practices. This research paves the way for future explorations, encouraging the continued evolution and application of biosensor technology in the pursuit of unraveling the complexities of the human brain and its capacity for language.

The analyzed studies employed a diverse range of biosensor technologies to investigate specific language areas within the brain. Researchers utilized tools such as the Magnes 2500 WH neuromagnetometer and microwire-based sensors to examine areas such as the receptive and expressive language cortex, Wernicke’s area, and the frontal cortex, aiming to map language function and activation patterns. Microwire-based biosensors were also employed to study language processing in rat hippocampal slices and freely moving rat brains, exploring real-time dynamics in both in vitro and in vivo settings. Further, innovative techniques such as magnetoresistive sensors incorporated into micromachined Si probes allowed for the detection of magnetic fields associated with language processing within neural tissue. These examples highlight the diverse approaches used to target and analyze specific brain regions involved in language.

Beyond focusing on specific brain regions, several studies investigated language function by analyzing broader brain activities and responses. For instance, researchers utilized a highly sensitive Magneto-Impedance (MI) sensor to measure alpha rhythms in the occipital region and Event-Related Field P300 in frontal, parietal, and temporal regions, aiming to capture real-time brain activity related to language processing. Similarly, studies employing neural biosensors and silicon microstructure biosensors focused on neuron-based activities, aiming to understand neurological events and drug responses related to language function. These approaches demonstrate a shift towards understanding language processing by analyzing broader neural activities and responses rather than focusing solely on specific brain regions.

## Figures and Tables

**Figure 1 diagnostics-14-01535-f001:**
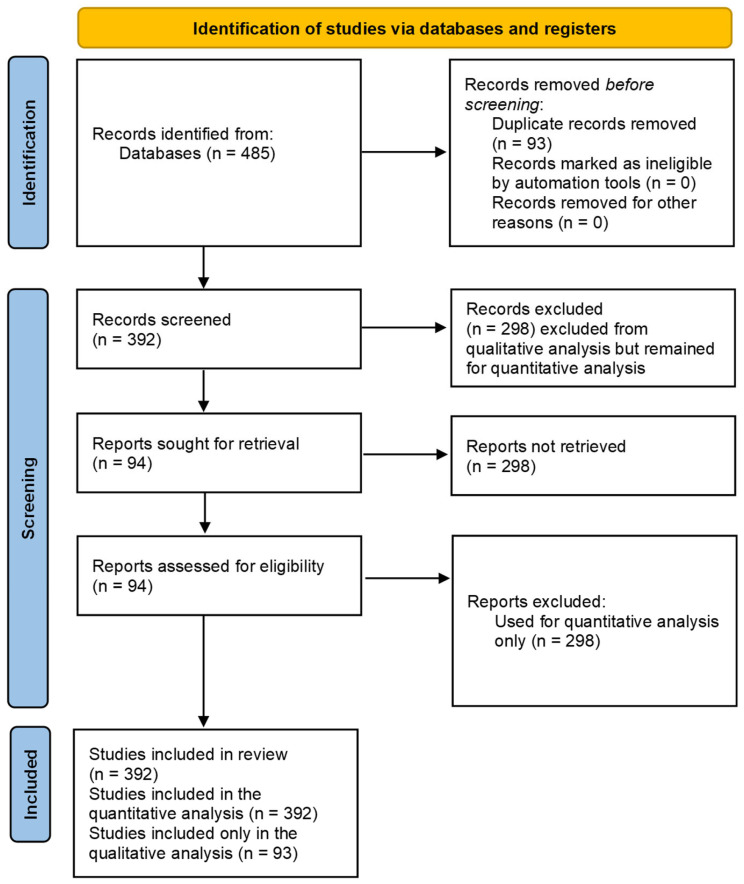
PRISMA Flow Diagram.

**Figure 2 diagnostics-14-01535-f002:**
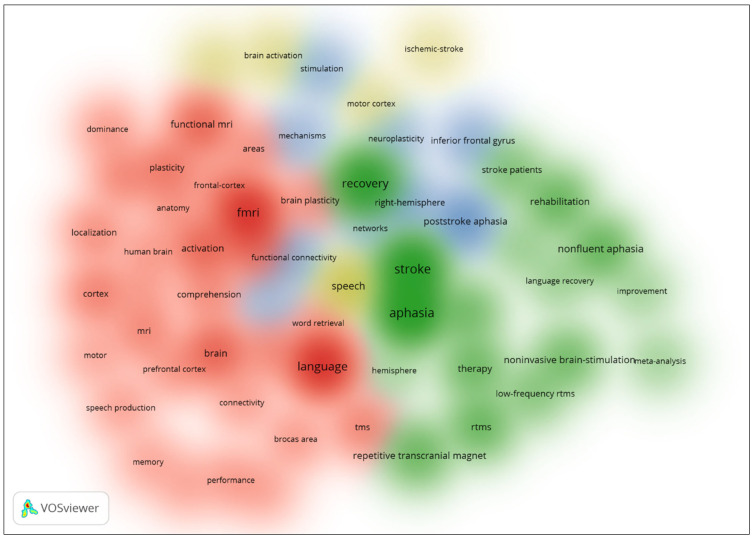
Density Visualization of Co-occurrence from Web of Science.

**Figure 3 diagnostics-14-01535-f003:**
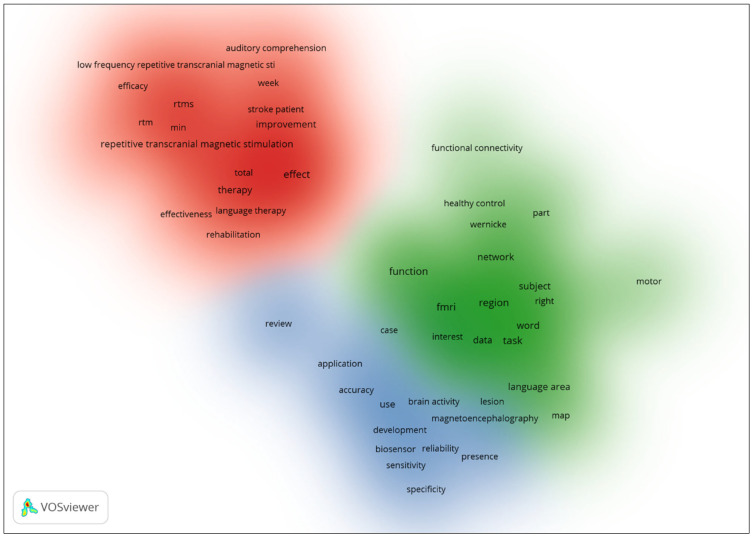
Density Visualization of Co-occurrence from Scopus and SciSpace.

**Figure 4 diagnostics-14-01535-f004:**
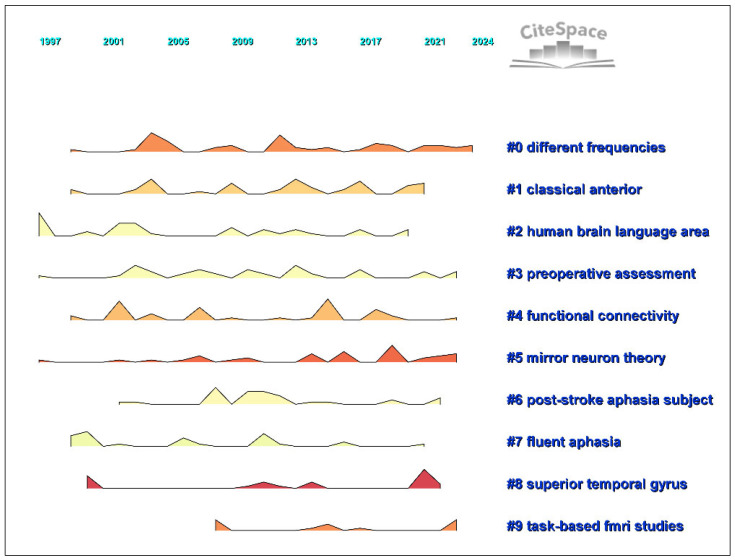
A Landscape Visualization of the Largest 10 Clusters.

**Figure 5 diagnostics-14-01535-f005:**
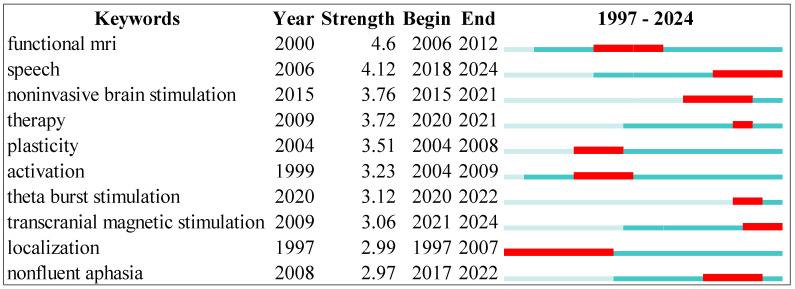
Top 15 Keywords with the Strongest Citation Bursts.

**Figure 6 diagnostics-14-01535-f006:**
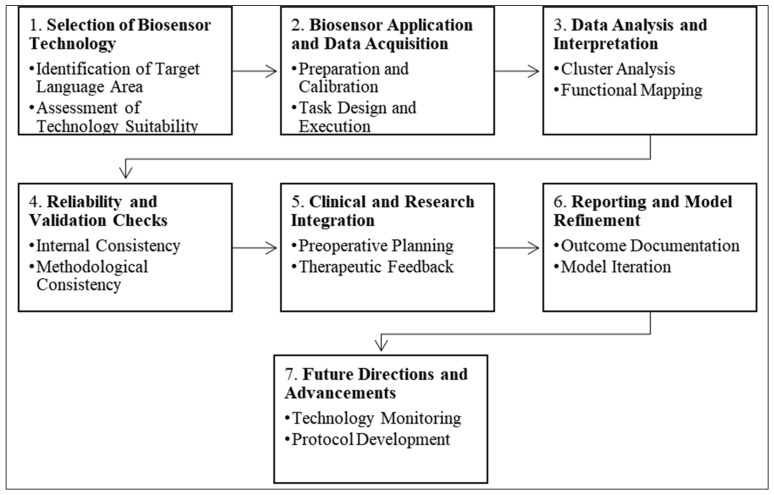
A Model Guiding the Use of Biosensors in Language Area Detection in the Brain.

**Table 1 diagnostics-14-01535-t001:** Search Strings Used to Retrieved Data.

Database	Query: Friday, 29 March 2024	Result
Scopus	(TITLE (“biosensor” OR “functional magnetic resonance imaging” OR “fmri” OR “cortico-cortical evoked potentials” OR “magnetoencephalography” OR “repetitive transcranial magnetic stimulation”) AND TITLE (“aphasia” OR “brain damage” OR “language areas”)) AND (LIMIT-TO (DOCTYPE, “ar”) OR LIMIT-TO (DOCTYPE, “re”) OR LIMIT-TO (DOCTYPE, “ch”))	168
Web of Science	“biosensor” or “Functional magnetic resonance imaging” or “fMRI” or “cortico-cortical evoked potentials” or “Magnetoencephalography” or “repetitive Transcranial Magnetic Stimulation” (Title) and “aphasia” or “brain damage” or “language areas” (Title) and Article or Review Article or Book Chapters (Document Types)	127
SciSpace	Biosensors and language areas in the brain	190

**Table 2 diagnostics-14-01535-t002:** Summary of the Largest 10 Clusters.

ID	Size	Silhouette	Label (LSI)	Label (LLR)	Label (MI)	Year
0	53	0.706	post-stroke aphasia	different frequencies	extended fMRI-guided anodal	2012
1	46	0.801	sign language production	classical anterior	sound shape properties	2012
2	44	0.903	fMRI study	human brain language area	moderate-severe nonfluent aphasia patient	2005
3	43	0.765	language area	preoperative assessment	constraint-induced language therapy	2010
4	41	0.873	functional connectivity	functional connectivity	attempted naming	2009
5	36	0.805	non-fluent aphasia	mirror neuron theory	attempted naming	2015
6	33	0.882	post-stroke aphasia subject	post-stroke aphasia subject	GRN mutation	2011
7	28	0.825	language area	fluent aphasia	attempted naming	2005
8	25	0.865	speech production	superior temporal gyrus	post-stroke aphasia	2013
9	15	0.945	following naming therapy	task-based fMRI studies	attempted naming	2015

**Table 3 diagnostics-14-01535-t003:** A Synthesis of 25 Studies on Utilizing Various Biosensor Technologies for Detecting and Mapping Language Areas Within the Brain.

No.	Authors	Used Biosensor	Methods Used	Language Area Examined	Is It Reliable?	Limitations
1	[6]	- Magnes 2500 WH neuromagnetometer from Biomagnetic Technologies, Inc. - Promold ear inserts for auditory stimulation from International Aquatic Trades, Inc.	- Receptive and expressive language cortex, Wernicke’s area, frontal cortex. - Activation through language tasks, MEG mapping, and cortical stimulation.	- MEG-derived maps remain stable and reliable over time. - MEG maps provide valid information on language-specific cortex location.	- Task involves other cognitive operations, not exclusively linguistic. - Different tasks may lead to slightly different maps in the future.	- MEG data collected in normative experiments with healthy volunteers - Reliability, validity, and topographical accuracy assessed in patients with Wada procedure
2	[29]	- Microwire-based biosensor - Shank-type biosensor coated with enzymes	- Language areas were examined using functional MR imaging and electrocortical stimulation. - Location of language areas varied among subjects and tasks.	- Functional MR imaging is considered a useful presurgical planning tool. - Sensitivity/specificity for identifying language areas ranged from 81%/53% to 92%/0%.	- Specificity of functional MR imaging for language areas was low. - Sensitivity varied based on distance between activated areas.	- Enzyme-based microelectrode array biosensors - Microwire-based biosensor with periodic insertion into cannula
3	[33]	- Microwire-based biosensor - Shank-type biosensor coated with enzymes	- Rat hippocampal slices and freely moving rat brain regions - In vitro and in vivo brain structures analyzed for language.	- Reliable due to improved selectivity and real-time measurements. - Real-time dynamics observed in vitro and in vivo experiments.	- Non-normally distributed peak concentration values. - Use of independent Wilcoxon test for comparison.	- Innovative techniques for spatial resolution and sensitivity improvement - Application of NV centers for biological electromagnetic field measurement
4	[4]	- Microwire-based biosensor - Shank-type biosensor coated with enzymes	- Left cerebral hemisphere - Frontal, temporal, and parietal lobes	- Average activation map proved reliable in split-half analysis. - Functional maps obtained from 30 right-handed subjects were reliable.	- Non-normally distributed peak concentration values. - Use of independent Wilcoxon test for comparison.	- FMRI used to identify language processing areas in the human brain. - Language activation task compared with control task involving non-linguistic sounds.
5	[24]	- Magnetoresistive sensors - Incorporated in micromachined Si probes	- Brain area: Neural tissue - Language element: Magnetoresistive sensors	- Reliable: System combines high sensitivity sensors on micromachined Si probes. - Reliable: Electrical and magnetic behavior of sensors verified in tests.	- Sensitivity to detect nT range magnetic fields. - Limited to detecting ionic currents in electrically active neurons.	- Si-etch-based micromachining process for neural probes - Incorporation of an array of magnetoresistive sensors on probes
6	[8]	- Chalcogenide glass fibers with tapered sensing zone - Detect metabolic anomalies in hepatic tissues for pathology studies	- Mid-infrared (MIR) range used for spectral analysis - Detection of metabolic anomalies in hepatic tissues through spectroscopy	- Reliable: Spectral differences reflect metabolic alterations in liver tissues. - Reliable: Results confirmed by histologic studies.	- Limited penetration depth of evanescent wave in sample - Fiber can be used only once for each experiment	- Evanescent wave spectroscopy - Transmission infrared spectroscopy
7	[26]	- Highly sensitive Magneto-Impedance (MI) sensor - Real-time brain activity measurement and signal processing system	- Occipital region measures alpha rhythm. - Frontal, parietal, and temporal regions measure Event-Related Field P300.	- Reliability confirmed by comparing results with relevant research. - MI sensor shows capabilities for brain activity measurement applications.	- MI sensor system requires signal amplification and noise reduction. - MI sensor system may have a limited frequency detection range.	- Real-time brain activity measurement using highly sensitive MI sensor - Signal processing system for monitoring brain activity in real-time
8	[25]	- Neural biosensor - Silicon microstructure biosensor	- Neuron-based devices for neurological research and drug development. - Collaboration between researchers from different disciplines for future progress.	- Yes, neuron-based devices can measure neurological events with sensitivity. - Potential for testing novel neuron-active drugs and fundamental research.	- Large difference in charge carrier mobilities between ionic and electronic conduction. - Understanding complex neurobiological responses and bioelectronic interface challenges.	- Measure inter-neuron contact, extracellular metabolic products, neuron-small molecule interactions - Evaluate neuron-active compounds, pharmacological activity, and neurological events with sensitivity
9	[1]	- Light LisNRs - Photosensitive MRI probe	- Light distribution in living rat brain studied using novel sensor. - Sensor detects light intensity in deep brain tissues effectively.	- Reliable: Steady performance in rat brain, consistent light response. - Potential for further optimization through adjustments.	- Limited ability to image signals in deep tissues - Conventional fluorescent sensors affected by absorption and scattering in tissues	- Fluorescent sensors including quantum dots, up conversion nanoparticles, and fluorescent proteins. - Novel sensor converts light into a magnetic signal for MRI.
10	[21]	- NTA-modified biosensor - Cyt c-based biosensor	- Brain area: Rat brain during ischemia and reperfusion - Language element: Cyt c-based O2- sensors suffer from interference	- High selectivity challenge due to coexisting biological species in the brain. - Difficult to elucidate mechanism due to lack of selective methods.	- Interference from reductants in the brain limits sensor application. - Lack of selective and reliable analytical methods for real-time determination.	- Electrochemical biosensors with high selectivity - ZnOSOD microelectrode for O2- determination in bean sprouts
11	[31]	- Quartz and silicon substrates with microelectrodes for impedance detection. - Quartz preferred over silicon due to reduced stray capacitances.	- Rat hippocampal slices and freely moving rat brain regions - In vitro and in vivo brain structures analyzed for language.	- Quartz substrate allows accurate measurement of sensor head capacitance. - Silicon substrate with grounding recovers expected sensor head capacitance value.	- High stray capacitances from conductive Si substrate affect biosensor accuracy. - Stray capacitance degrades resolution in biosensor readout.	- Direct experimental comparison of 10 μm disk electrodes - Analysis of accuracy and resolution in impedance detection
12	[28]	- Multisensory Microelectrode Biosensor - MEMS biosensor	- Frontal cortex: responsible for complex thinking, decision making, social behaviors. - Granule cells: studied in dense knot deep in the brain.	- Yes, PEGDE method shows over 90% positive results. - PEGDE used for specific neurotransmitter detection in CNS.	- Increased internal traffic and potential brain damage - Separate sensors for each brain parameter causing inefficiency	- PEGDE method for neuron identification - Multisensory microelectrode biosensor for neuron activity detection
13	[10]	- Glutaraldehyde and PEGDE-based biosensors were used. - Glutaraldehyde biosensors overestimated glutamate levels compared to PEGDE biosensors.	- Brain interstitial fluid for neurotransmitter detection - Glutamate and glucose oxidase specificity in biosensors	- Glutaraldehyde-based biosensors overestimated glutamate levels compared to capillary electrophoresis. - PEGDE-based biosensors showed consistent glutamate levels with capillary electrophoresis.	- Glutaraldehyde decreased substrate specificity, overestimating glutamate levels. - PEGDE maintained substrate specificity, providing accurate glutamate detection.	- Glutaraldehyde and PEGDE used for enzyme immobilization in biosensors. - HPLC and CE-LIF used for confirming biosensor specificity.
14	[19]	- Functional magnetic resonance imaging - Positron emission tomography	- Primary motor cortex (M1) - Secondary motor areas such as PMd and supplementary motor area	- Yes, based on non-invasive brain function study techniques. - Mechanisms identified led to novel motor rehabilitation approaches.	- Weakness: Lack of universally accepted treatment for stroke. - Measure: Non-invasive techniques such as fMRI, PET, TMS, EEG, and MEG used.	- Functional neuroimaging studies - Integration of behavioral, computational, and neurophysiological approaches
15	[30]	- Field-effect transistor-based biosensors (bioFETs) - Micropurification chip (MPC) for cancer marker detection.	- Charge-based biosensors and their limitations in detection performance. - Strategies to improve sensitivity and signal-to-noise ratio in biosensors.	- Reliability and accuracy are paramount for clinical translation. - Intrinsic device noise and screening by electrolyte limit performance.	- Intrinsic device noise limits smallest measurable signal. - Screening by electrolyte environment reduces measurable signal.	- Signal-to-noise ratio as a universal performance metric. - Alternative functionalization and detection schemes for physiological conditions.
16	[7]	- Microwire-based biosensor - Shank-type biosensor coated with enzymes	- Rat hippocampal slices and freely moving rat brain regions - In vitro and in vivo brain structures analyzed for language.	- Reliable due to improved selectivity and real-time measurements. - Real-time dynamics observed in vitro and in vivo experiments.	- Non-normally distributed peak concentration values. - Use of independent Wilcoxon test for comparison.	- Magnetoencephalography (MEG) - Event-related magnetic fields (ERFs) recorded for language tasks.
17	[32]	- Microwire-based biosensor - Shank-type biosensor coated with enzymes	- Rat hippocampal slices and freely moving rat brain regions - In vitro and in vivo brain structures analyzed for language.	- Reliable due to improved selectivity and real-time measurements. - Real-time dynamics observed in vitro and in vivo experiments.	- Non-normally distributed peak concentration values. - Use of independent Wilcoxon test for comparison.	- Enzyme-based microelectrode array biosensors - Microwire-based biosensor with periodic insertion into cannula
18	[20]	- Microwire-based biosensor - Shank-type biosensor coated with enzymes	- Rat hippocampal slices and freely moving rat brain regions - In vitro and in vivo brain structures analyzed for language.	- Reliable due to improved selectivity and real-time measurements. - Real-time dynamics observed in vitro and in vivo experiments.	- Non-normally distributed peak concentration values. - Use of independent Wilcoxon test for comparison.	- PtGOx-PPD sensors - Incorporating enzyme in monomer solution
19	[9]	- Enzyme nanoparticle-based biosensors - Cell-based biosensors with immobilized cells and tissues	- Bioaffinity and biocatalytic devices - Cell-based biosensors, enzyme immunosensors, DNA biosensors	- Biosensors meet criteria for sensitive, accurate diagnostic tools. - Nanotechnology optimizes biochips for precise bedside monitoring.		- Amperometric, potentiometric, or optical transducers - Enzyme nanoparticle-based biosensors using nanotechnology
20	[11]	- Electrochemical sensors - Multiple analytes	- Biological elements: enzymes, antibodies, micro-organisms, tissues, organelles - Transduction elements: mass-based, electrochemical, optical biosensors	- Biosensors are reliable due to high sensitivity and selectivity. - Stability, reproducibility, and low cost are important for reliability.	- Stability of biological component outside normal environment is critical. - Matching appropriate biological and electronic components is essential for relevance.	- Transducer technologies: Specific for biological sensor interaction. - Enzyme biosensors: Detect organophosphates and carbamates in pesticides.
21	[5]	- Microdialysis probe for sampling extracellular brain fluid - Fast-scan cyclic voltammetry (FSCV) with carbon fiber microelectrodes	- pH levels and acidosis patterns in brain tissue examined - Time course of insertion-related bleeding and coagulation studied	- Large performance variability observed due to tissue damage - Insertion injury initiates inflammatory tissue response impacting sensor performance	- Chronic tissue response decreases signal sensitivity over time. - Pro-inflammatory molecules adsorb onto implant surface, perpetuating inflammation response.	- Microelectrodes for fast-scan cyclic voltammetry (FSCV) and electrophysiology - Microdialysis probes for sampling and detecting various neurochemicals
22	[22]	- Grip force sensor - Tool for measuring language-induced activity in motor structures	- Motor brain structures activated by action words - Grip force sensor measures language-induced motor activity	- Grip force sensor provides accurate data on motor brain activity. - Continuous monitoring allows fine-grained estimation of motor activity.	- Idiosyncratic grip force signatures present in some participants. - Noise in data of some participants affecting correct exploitation.	- Grip force sensor for online measurement of language-induced activity - Data recording and stimulus presentation on two distinct computers
23	[12]	- Glutamate oxidase for central nervous system neurotransmitter detection. - Three-enzyme system for optimal acetylcholine detection.	- pMTG, opIFG, and vPrG involved in word processing during naming. - No response errors mainly located in trIFG, vPrG, and mMTG.	- Object naming task is the most discriminative for language mapping. - rTMS can identify different brain areas for each language task.	- No measurement of naming, reading, or generation latencies. - Hesitation errors only compared to baseline testing, introducing subjectivity.	- 19 subjects performed language tasks during rTMS mapping - 5 Hz/10 pulses applied with 0 ms delay
24	[27]	- Central neuro-structures - CCZ	- Medulla oblongata - Central chemoreceptor zone (CCZ)	- Results discussed with prospects for using chemical biosensors. - Error range in measurements was 5–8.	- Lack of discussion on potential measurement errors. - Limited information on calibration procedures for accuracy assessment.	- Spherical drop and planar source methods for measuring diffusion coefficient - Planar diffusion method involving changes in C0 or distance R
25	[23]	- Genetically encoded fluorescent sensors - New bioluminescent sensors for deep-tissue imaging	- Brain activities, neuronal imaging using fluorescent protein-based biosensors. - Protein engineering, optimizations, and experimental applications for imaging brain activities.	- Reliability is influenced by protein engineering efforts and experimental applications. - Future developments aim to enhance reliability and fill technological gaps.	- Factors influencing sensor performance analyzed through protein engineering efforts. - Future developments can fill technological gaps for improved sensor performance.	- Genetically encoded fluorescent sensors - New bioluminescent sensors for deep-tissue imaging

**Table 4 diagnostics-14-01535-t004:** A Synthesis of Key Studies on Biosensor Applications in the Detection and Mapping of Language Areas in the Brain.

No.	Citation	Aim	Method	Biosensors Implication
1	[34]	To examine if language area connectivity is asymmetric in the brains of normal children.	Functional connectivity analysis using MATLAB CONN toolbox	Suggests biosensors can detect lateralization in language areas, important for understanding development.
2	[4]	Identify potential language processing areas in the human brain using a broad definition of language.	Functional MRI with phonetic and semantic analysis tasks	Demonstrates the precision of biosensors in mapping language processing networks in the brain.
3	[35]	Assess the neurobiological impact of aphasia treatment including speech therapy and the role of hemispheres.	Functional imaging and rTMS	Highlights the potential of biosensors to monitor and enhance the effectiveness of speech therapy in aphasia.
4	[36]	Investigate connectivity in the reorganized language network of temporal lobe epilepsy patients.	CCEP study	Indicates biosensors can reveal functional shifts in language networks due to epilepsy.
5	[37]	Determine the reliability of preoperative language fMRI for patients with brain tumors.	Review of studies comparing language fMRI with direct cortical stimulation (DCS)	Raises questions about the reliability of biosensors for preoperative language area mapping in tumor-related cases.
6	[38]	Explore rTMS as a treatment for post-stroke aphasia and its challenges.	rTMS	Suggests biosensors could be essential in optimizing rTMS for language function improvement post-stroke.
7	[39]	Evaluate fMRI and ECS effectiveness in localizing language functional areas during awake craniotomy.	Functional MRI and electrical cortical stimulation (ECS)	Supports the use of biosensors in enhancing the accuracy of language area localization during surgery.
8	[40]	Explore rTMS as a language improvement treatment in a patient with chronic crossed aphasia.	Pre- and post-rTMS fMRI for noun generation and sentence completion	Demonstrates the potential of biosensors in monitoring and enhancing rTMS treatment efficacy for aphasia.
9	[41]	Establish a non-invasive method combining fMRI and MEG for identifying language areas.	Combined fMRI and MEG, validated against the Wada test	Shows biosensors’ reliability in non-invasively identifying language areas, offering an alternative to invasive tests.
10	[42]	Evaluate the effectiveness of rTMS therapy for post-stroke non-fluent aphasia.	Literature review and analysis of rTMS studies	Suggests biosensors are key in assessing rTMS as a viable treatment for post-stroke language impairments.
11	[43]	Examine the cerebral organization for sign language production using fMRI.	4 T fMRI in deaf native ASL users and hearing controls performing a naming task.	Indicates biosensors can capture the neural correlates of language production across different modalities.
12	[44]	Analyze the efficacy of rTMS in treating stroke-induced aphasia.	Systematic review and meta-analysis of rTMS clinical studies.	Points to biosensors’ role in establishing effective rTMS parameters for aphasia treatment.
13	[45]	Test neuronavigation-guided rTMS for aphasia and compare its precision to conventional methods.	Neuronavigation-guided rTMS versus conventional TMS with picture naming task	Highlights biosensors’ precision in targeting and potentially improving outcomes in aphasia therapy.
14	[46]	Investigate the reorganization of language areas in patients with CNS tumors using fMRI.	fMRI prior to surgical treatment in patients with tumors near language centers	Suggests biosensors can detect functional rearrangements in language areas due to CNS tumors.
15	[47]	Assess the reliability of 3-T fMRI for localizing language-related function by comparing with ECS.	Detailed analysis comparing 3-T fMRI results with extraoperative ECS mapping	Reveals variations in biosensor reliability for language area localization, dependent on brain regions.
16	[48]	Evaluate the effect of personalized rTMS in patients with primary progressive aphasia.	Randomized, double-blind pilot study with active-versus control-site rTMS	Demonstrates biosensors’ role in personalizing rTMS to enhance language and cognitive functions in PPA.
17	[49]	Analyse the usefulness of preoperative language fMRI by correlating with intraoperative cortical stimulation results.	Correlation of fMRI data with intraoperative direct cortical stimulation (DCS) findings	Challenges the sole reliance on biosensors for critical surgical decisions, advocating for multiple modalities.
18	[50]	Explore the efficacy of fMRI-guided rTMS for chronic post-stroke aphasia.	fMRI-guided rTMS using an excitatory stimulation protocol	Suggests biosensors can contribute to language recovery by guiding rTMS to target areas.
19	[51]	Discuss fMRI’s role in assessing language areas in children with non-lesion focal epilepsy.	Application of fMRI in pre-surgical planning for epileptic patients	Demonstrates biosensors’ utility in non-invasively mapping language areas for surgical planning.

**Table 5 diagnostics-14-01535-t005:** A Synthesis of Top-Cited Studies on Biosensor Applications in the Detection and Mapping of Language Areas in the Brain.

No.	Authors	Aim	Method	Biosensors Implication	Citations
1	[4]	To identify brain regions involved in phonological and lexical-semantic language processing.	fMRI with phonetic and semantic analysis tasks.	Demonstrates the capability of fMRI to delineate language processing areas, potentially refining classical language models.	1043
2	[49]	Examine the utility of preoperative fMRI correlated with intraoperative cortical stimulation in patients with brain tumors.	fMRI with naming and verb generation tasks, correlated with direct cortical stimulation.	Indicates the need for multimodal approaches to improve the accuracy of preoperative language mapping with biosensors.	299
3	[37]	Assess the reliability of language fMRI compared with direct cortical stimulation in brain tumor surgery.	Review of studies comparing preoperative language fMRI with intraoperative direct cortical stimulation.	Highlights the variability in biosensor reliability, suggesting the need for methodologically robust studies for validation.	256
4	[52]	Determine the role of the right inferior frontal gyrus in language function recovery post-stroke.	rTMS combined with PET during a semantic task.	Provides evidence of the right hemisphere’s role in language function post-stroke, suggesting a compensatory mechanism detectable by biosensors.	246
5	[53]	Analyze the neural basis of conduction aphasia and its relation to phonological short-term memory.	Aggregate analysis of lesion and fMRI data.	Supports the role of the left temporoparietal region in language processing, with implications for biosensor-based diagnoses.	209
6	[54]	Develop an fMRI protocol for localizing critical language areas as an alternative to intraoperative mapping.	fMRI with language tasks compared to intraoperative electrocortical stimulation mapping.	Suggests that fMRI can reliably predict the presence or absence of language function, with potential to guide surgical decisions.	203
7	[55]	Map language areas in the anterior temporal lobe using fMRI to assist in epilepsy surgery.	Multicenter normative fMRI study comparing narrative comprehension with arithmetic tasks.	Highlights efficacy of biosensors in pre-surgical mapping to minimize language and memory deficits post-surgery.	179
8	[56]	Investigate disruptions in reading-related brain connectivity in children with dyslexia.	fMRI connectivity analysis during a continuous reading task.	Points to the potential of biosensors to unravel the neural underpinnings of dyslexia and inform interventions.	169
9	[2]	To investigate the neural mechanisms of word retrieval in aphasic patients.	fMRI during covert word retrieval tasks in aphasic patients.	Provides insights into the neural adaptations that underpin language recovery in aphasia.	130
10	[57]	To evaluate the impact of rTMS on naming in nonfluent aphasia patients.	Overt naming fMRI before and after a series of rTMS treatments.	Suggests that biosensors can track changes in language processing following rTMS, offering potential therapeutic insights.	126

## Data Availability

The raw data supporting the conclusions of this article will be made available by the authors on request.

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
