# Peer review of "Using Biosensors to Detect and Map Language Areas in the Brain for Individuals with Traumatic Brain Injury"

_diagnostics, 2024, doi:10.3390/diagnostics14141535_

Round 1
Reviewer 1 Report
Comments and Suggestions for Authors
Impressive manuscript. Please explain silhouettes and bursts.
Author Response
Dear Colleague,
Thank you so much for your effort and time on reviewing our paper. We are glad that everything sounded well for you.
Kind regards,
Authors
Reviewer 2 Report
Comments and Suggestions for Authors
The paper is an excellent project and very well written. I would suggest acceptance in its present form.
Author Response

(The authors gave the same response as above.)

Reviewer 3 Report
Comments and Suggestions for Authors
The article "Using Biosensors to Detect and Map Language Areas in 2 the Brain for Individuals with Traumatic Brain Injury " offers a new original and comprehensive view on the use of biosensors in neurolinguistics. The authors propose a new conceptual model for the use of biosensors in scientific research and clinical practice. This is an interesting and practically significant study, carried out with great care and a high level of detail.
The authors proposed a non-standard design of the Introduction section, where they simultaneously presented complex and complex information justifying the main objectives of the work. The authors quite correctly divided this section of the work into thematic sections, in which they presented in detail and convincingly the theoretical foundations supported by research on complex versatile aspects of this scientific and practical direction. This section is very convincing and generally gives a clear idea of the main purpose of the work.
In the methodological part of the work, the authors describe in detail and clearly the main methodological approaches that guided them in conducting the study. This section clearly and distinctly reflects the design of the work carried out and does not raise any questions.
In the Results section, the authors conducted two types of analysis according to which they made reasonable and convincing conclusions. Thus, in section 3.11, the authors provide a clear and reasoned justification that the use of biosensors promises to increase the effectiveness of preoperative assessment, minimizing postoperative speech deficit, as well as to offer a cost-effective alternative to MRI technology. Other findings in this section also reasonably show the involvement of biosensors in the early detection and development of targeted intervention strategies, optimizing the results of rehabilitation of patients with aphasia. The remaining conclusions of this section are also convincing and scientifically sound
The next section of the results is devoted to thematic analysis. In general, Table 3 looks rather cumbersome, and the authors should think about how to make this section of the work more compact and readable. However, the benefits of this section of the results are very obvious, so it is necessary to make Table 3 more compact.
In general, the Results section is well thought out and a large amount of data is optimized and presented in the form of tables and graphical diagrams, which in all cases are accompanied by a qualified and comprehensive and summarizing, as well as in some cases an explanatory conclusion, each of which makes these sections understandable and informative.
The Discussion section is quite specific and focused around the results obtained. This section, like all the previous ones, is well structured and contains constructive and important information about biosensors. The authors also included in the discussion sections limitations and consequences, which also contain objective and important information limiting the use of biosensors.
In the Conclusion section, however, it is necessary to more clearly indicate the main life hacks of the authors and show to what extent the material analyzed by them contributed to the identification and assessment of language areas of the brain, especially in people who have suffered traumatic brain injuries.
Comments on the Quality of English LanguageMinor stylistic adjustments are recommended
Author Response
Dear Colleague,
We do highly appreciate your effort on reviewing our paper. We have done our best to approach your concerns and suggestions. Please find below responses to your concerns. We also applied changes accordingly in the revised version of the manuscript. These are coloured in blue for your reference.
Once again, thank you so much.
Kind regards,
Authors
The article "Using Biosensors to Detect and Map Language Areas in 2 the Brain for Individuals with Traumatic Brain Injury " offers a new original and comprehensive view on the use of biosensors in neurolinguistics. The authors propose a new conceptual model for the use of biosensors in scientific research and clinical practice. This is an interesting and practically significant study, carried out with great care and a high level of detail.
Thank you.
The authors proposed a non-standard design of the Introduction section, where they simultaneously presented complex and complex information justifying the main objectives of the work. The authors quite correctly divided this section of the work into thematic sections, in which they presented in detail and convincingly the theoretical foundations supported by research on complex versatile aspects of this scientific and practical direction. This section is very convincing and generally gives a clear idea of the main purpose of the work.
Thank you.
In the methodological part of the work, the authors describe in detail and clearly the main methodological approaches that guided them in conducting the study. This section clearly and distinctly reflects the design of the work carried out and does not raise any questions.
Thank you.
In the Results section, the authors conducted two types of analysis according to which they made reasonable and convincing conclusions. Thus, in section 3.11, the authors provide a clear and reasoned justification that the use of biosensors promises to increase the effectiveness of preoperative assessment, minimizing postoperative speech deficit, as well as to offer a cost-effective alternative to MRI technology. Other findings in this section also reasonably show the involvement of biosensors in the early detection and development of targeted intervention strategies, optimizing the results of rehabilitation of patients with aphasia. The remaining conclusions of this section are also convincing and scientifically sound
Thank you so much.
The next section of the results is devoted to thematic analysis. In general, Table 3 looks rather cumbersome, and the authors should think about how to make this section of the work more compact and readable. However, the benefits of this section of the results are very obvious, so it is necessary to make Table 3 more compact.
We are sorry about Table 3 format and intensity of content. However, please note that this is due to the orientation of the layout. While we submitted all the able with landscape layout orientation, the production team has changed this to portrait orientation layout according to the journal requirements. We will pass this note them and say if they will agree to bring them back to the original format.
In general, the Results section is well thought out and a large amount of data is optimized and presented in the form of tables and graphical diagrams, which in all cases are accompanied by a qualified and comprehensive and summarizing, as well as in some cases an explanatory conclusion, each of which makes these sections understandable and informative.
Thank you.
The Discussion section is quite specific and focused around the results obtained. This section, like all the previous ones, is well structured and contains constructive and important information about biosensors. The authors also included in the discussion sections limitations and consequences, which also contain objective and important information limiting the use of biosensors.
Thank you.
In the Conclusion section, however, it is necessary to more clearly indicate the main life hacks of the authors and show to what extent the material analyzed by them contributed to the identification and assessment of language areas of the brain, especially in people who have suffered traumatic brain injuries.
Thank you. We expanded the conclusion adding to paragraphs in response to this.
Comments on the Quality of English Language
Minor stylistic adjustments are recommended
Thank you. We went through the whole manuscript again and applied comprehensive editing including spelling, grammar, language accuracy and readability. We leave the resting formatting issue for the production team.
Once more, thank you so much for all your feedback.
Authors